# Performance Evaluation of Phase Change Materials to Reduce the Cooling Load of Buildings in a Tropical Climate

Punita Sangwan [1], Hooman Mehdizadeh-Rad [1,2], Anne Wai Man Ng [1,2], Muhammad Atiq Ur Rehman Tariq [3,4] and Raphael Chukwuka Nnachi [5,*]

1 College of Engineering, Information Technology and Environment, Charles Darwin University, Ellengowan Drive, Brinkin, NT 0810, Australia; punita@students.cdu.edu.au (P.S.); hooman.mehdizadehrad@cdu.edu.au (H.M.-R.); anne.ng@cdu.edu.au (A.W.M.N.)
2 Energy and Resources Institute, Charles Darwin University, Darwin, Ellengowan Drive, Brinkin, NT 0810, Australia
3 College of Engineering and Science, Victoria University, Melbourne, VIC 8001, Australia; atiq.tariq@yahoo.com
4 Institute for Sustainable Industries & Liveable Cities, Victoria University, P.O. Box 14428, Melbourne, VIC 8001, Australia
5 Faculty of Biological Sciences, Alex Ekwueme Federal University, Ndufu Alike Ikwo, Abakaliki P.M.B 1010, Nigeria
* Correspondence: nnachi.raphael@funai.edu.ng

**Abstract:** Tropical region such as Darwin has similar weather patterns throughout the year, thus creating higher energy demands in residential buildings. Typically, buildings consume about 40 per cent of the total energy consumption for indoor heating and cooling. Therefore, building envelopes are linked with design strategies such as the use of thermal energy storage and phase change materials (PCM) to minimize this energy consumption by storing a large amount of thermal energy. Primarily, PCMs are targeted by researchers for use in different components of buildings for thermal efficiency; thus, this study aimed to provide a suitable PCM to optimize indoor thermal comfort and minimize the cooling loads of residential buildings in tropical climates through simulation of a tropical climate building and provide optimum thickness for the selected material. Microencapsulated PCM mixed with gypsum in wallboards were used to reduce the cooling load of a building located in Darwin. The cooling load of the building was calculated using Revit software. A comparison of the cooling load of the building was carried out using PCM-incorporated wallboards of thicknesses of 0 cm, 1 cm and 2 cm in Energy Plus software. The total cooling load decreased by 1.1% when the 1-centimetre-thickness was applied to the wall, whereas a 1.5% reduction was obtained when a 2-centimetre-thick PCM layer was applied. Furthermore, the reduced cooling loads due to impregnation of the PCM-based gypsum wallboard gave reduced energy consumption. Ultimately, the 2-centimetre-thickness PCM-based gypsum wallboard gave a maximum reduction in cooling load with a 7.6% reduction in total site energy and 4.76% energy saving in USD/m$^2$/year.

**Keywords:** thermal comfort; cooling loads; simulation; tropical climates; phase change materials; Energy Plus; Revit

## 1. Introduction

For the last three decades, the energy efficiency of buildings has been getting much attention due to the substantial energy demands of maintaining indoor thermal comfort in modern buildings [1,2]. Generally, it is critical to construct energy-efficient buildings as the increased demand for fossil fuel creates environmental problems like climate change and an increase in the cost of fossil materials [3]. Previously, buildings were constructed with wider walls that could store a large amount of sensible heat and provide natural conditioning by controlling the temperature variations [4]. However, in modern buildings,

the wall's thermal mass has been reduced to save materials, time and transportation [5]. Ultimately, the lightweight buildings are getting huge temperature variations because of this new development, which leads to low thermal storage and extreme internal cooling and heating loads [6,7]. Therefore, it is crucial to design and implement efficient energy-saving techniques in buildings. Building envelopes such as walls, roofs, windows and floors isolate the indoor environment from the outside weather conditions to maintain indoor thermal comfort [8]. Therefore, building envelopes present an opportunity to enhance indoor thermal comfort and reduce energy consumption by influencing the heating and cooling loads of a building [9]. Many steps have been taken by Heat, ventilation and air condition (HVAC) engineers, such as incorporating facades, hydronic systems, thermal conditioning by radiation, thermal storage devices and many more to reduce energy consumption in buildings [10]. To ensure the thermal comfort of a building, it is evident that cooling and heating loads will be affected, but recent development in construction materials has changed this phenomenon with the significant use of phase change materials (PCM) [11,12].

## 2. Phase Change Materials

PCM provides additional thermal mass to a building's envelope to maintain indoor temperature fluctuations by absorbing and releasing heat during the material's phase change [13]. These thermal energy storage materials help maintain the indoor ambient temperature within a specific range close to the transition temperature of the PCM [14]. When PCM attain their melting temperature during the daytime, the indoor ambient temperature increases and the chemical bond in the material breaks (As illustrated in Figure 1A), causing additional heat to be absorbed by the material and its state changing from solid to liquid [15]. Similarly, when the temperature drops below the PCM's freezing temperature at night, the PCM release energy and its state changes from liquid to solid [16]. If the optimal selection of phase melting and freezing temperature matches, the required indoor comfort temperature can help in reducing the cooling loads by absorbing the extra heat [15].

PCM is categorized as organic, inorganic and eutectic [17]. The subcategories of PCM such as paraffin are the most frequently used irrespective of the region, and the maximum frequency of usage is up to 87.5%. This could be because the mixtures of paraffin in different mass proportions have a more comprehensive phase change temperature range and higher phase change latent heat (as illustrated in Figure 1B) [18]. Thus, paraffin mixtures can be used in different thermal storage fields through the modification of the mixed proportion [19].

Paraffin PCM was chosen for this study because they are readily available, inexpensive and melt at different temperatures relating to their carbon chain length with the general formula CnH2n+2 [20]. Their melting temperature and phase change enthalpy increase with the length of the carbon chain. When the number of carbon atoms in the paraffin molecule is between 13 and 28 [20,21], the melting temperature falls within approximately $-5$ to 60 °C, a temperature range covering building applications in most climates around the world [22]. They are chemically stable; their volume increase by 10% upon melting, which is similar to many inorganic materials but less critical as paraffin are softer and therefore build up smaller forces upon expansion [23]. Paraffin is safe and non-reactive; it does not react with the most common chemical reagents [24]. The latent heat stored to gypsum wallboard during melting and freezing depends on phase change temperature, location, the thickness of the PCM wallboard layer, the shape of the enthalpy curve and the temperature range of phase change [25].

In a hot climate, the indoor temperature changes quickly because of higher temperatures outside buildings [15]. In such cases, PCM can be most effective as they have high volumetric heat capacity at the time of phase transition, and it could be 30 times higher than that of concrete or other massive construction materials [5]. The melting temperature, thermal conductivity and Thermal Energy Storage (TES) density are the best measures to test the thermal performance of PCM's integration into buildings [26]. PCM with rapid

melting and crystallization/solidification points are suitable for TES applications [3]. How-ever, designing an efficient heat storage system is challenging because it is not restricted to the system's thermal performance but also involves costs, safety, and sustainability of materials used and processes employed [3].

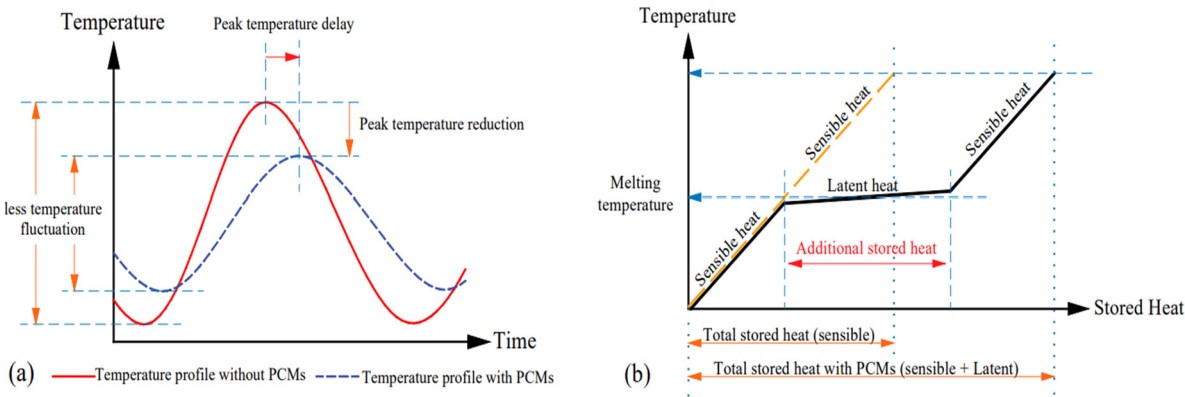

**Figure 1.** (**a**) Comparison of a building's indoor temperature profile with and without PCMs. (**b**) Comparison of latent heat & sensible heat in material storage [27].

### 2.1. Heat Transfer through a Wall

The total energy utilised by a household is consumed through heating of water, refrigeration, lightening, charging of electronic devices and gadgets. This energy that is consumed through heating and cooling is largely dependent on the type of house, its location and mode of construction. This heat transfer through a building can occur either by conduction, convention or radiation. Heat transfer through building structures such as walls and floor occur whenever a difference exists between the conditioned indoor space of a building and the surrounding outdoor temperature [28,29]. To determine the heat transfer through a building, the indoor conditions are assumed to be constant, although it also varies with climates and seasons. For instance, in cold countries and during the winter period, heating load calculations are dependent upon peak or near peak conditions that takes place shortly before sunrise, and there is little or no difference in the surrounding outdoor temperature in cold countries during winter [30,31].

In terms of heat transfer, walls are the most prominent source of heat transfer in a building. Generally, walls are exposed to the sun; therefore, calculating the temperature difference is critical. For calculating heat flow, we need to get the area of the wall excluding the area of windows, the transmission coefficient for external walls and the mass flow rate per unit from the ASHRAE tables (ASHRAE, Inc., Peachtree Corners, GA, USA, (n.d.)). Some important parameters should be considered to calculate the heat transfer through walls, for instance, the equivalent temperature difference when the wall is in shade or exposed to the sun, maximum solar heat gain at the direction of the wall facing for a desired month and latitude, wall facing towards north/south/east/west/direction.

The first step in estimating the heat transfer through a wall is to determine the design heat transmission coefficient. Thus, in terms of thermal efficiency, solar and transmission gain through walls can be calculated from the given equation below:

$$Q = U \times A \times (CLTD) \tag{1}$$

Q = Cooling load (Btu/hr = 0.293 watts);
U = overall heat transfer coefficient (Btu/hr-ft$^2$-F = 5.6783 W/(m$^2$·K);
A = Area (Ft$^2$ = 0.093 m$^2$);
CLTD = Equivalent temperature difference (F = 255.928 K).

Here, Q describes 'Sensible heat flow' that affects Heat Ventilation & Air Conditioning equipment size and energy consumption, and the goal is to minimize this value for overall

economics and efficiency. Higher Q value imposes high first- and recurring operation costs on the HVAC system [32]. A is a function of a building's form. The area values are computed from building plans and elevations drawings. the U-value describes the rate of heat flow through a building element. It is the reciprocal R-value (U = 1/R), where R is the total resistance of the materials used in constructing the wall. The higher the R-value, the higher the insulating value of the material or the lower the U-value, the higher the insulation value of the material. Energy efficiency standards set a maximum U-factor value and are calculated from the material information provided in building drawings. The cooling load temperature difference (CLTD) value is derived from the ASHRAE table for any wall, rood, latitude and hour of the day [32]. CLTD is a theoretical temperature difference that accounts for the combined effects of inside and outside air temp difference, daily temp range, solar radiation and heat storage in the construction assembly/building mass. It is affected by orientation, tilt, month, day, hour, latitude, etc. CLTD factors are used for adjustment to conductive heat gains from walls, roof, floor and glass [28].

### 2.2. Cooling Load Principle

To maintain a uniform thermal condition, the cooling load principle must be applied. The cooling load is the rate at which a cooling system or process must remove heat from a conditioned zone to maintain constant dry bulb temperature and humidity [33]. The building components which affect the cooling loads are external input (walls, roof, floor, window and ceiling), internal supply or internal load (lightening, occupants, appliances and equipment), infiltration (air leaks, moisture migration), system (duct leakage, heating, ventilation) [32]. Cooling load calculation methodologies are based on heat transfer by conduction, convection, radiation [33] and include heat balance, radiant time series, cooling load temperature difference [33]. Calculation of thermal loads of buildings is altered for cooling in summer for the accuracy of design and appropriate equipment or materials to adapt the thermal comfort and reduce the cooling loads in the selected place.

For standard cooling load calculations, many pieces of software have been designed by HVAC engineers, and the most popular ones include HAP, REVIT and ENERGY PLUS. This software examines each space of a building from the months and daytime. For example, walls are expected to provide thermal comfort within the building and at the same time, the thermal resistance (R-value) of the wall is a critical parameter and highly influences the energy consumption and increases the cooling loads [34], especially in the buildings where the ratio between the wall and total envelope area is more significant [35]. Based on the construction materials, walls are classified as wood-based, metal-based, or masonry-based.

To estimate the cooling load of an area or building, four methods have been suggested, including cooling Load by Transfer Function Method (TEM), Total Equivalent Temperature Difference (TETD) method, Cooling Load Temperature Difference (CLTD) method and Transfer Function Method (TFM). In this study, we adopted the CLTD method. This method involves the temperature difference in the case of building components (walls and roofs) and the cooling load factors in the case of solar heat gain through windows and internal heat sources. The following formula is used to estimate cooling loads using the CLTD method

$$Q = U \times A \times CLTDC$$

where
    Q is the net room conduction heat gain through roof, wall or glass (W);
    A is the area of the roof, wall or glass (m$^2$);
    U is the overall heat transfer coefficient (kW/m$^2 \cdot$K);
    CLTDC is the cooling load temperature difference (°C).

### 2.3. Thermal Comfort

Thermal comfort is a condition of the mind which expresses satisfaction with the thermal environment and is mainly affected by environmental factors such as air temperature, radiant temperature, air velocity and humidity. Thermal comfort also depends on the

cooling loads (the amount of heat energy that must be removed from an environment to maintain an optimum temperature) and the heating load (the amount of heat that must be added to an environment to ensure optimum temperature) of a place.

The metabolic rate for specific activity levels and clothing affects our thermal comfort as well. The Predicted Mean Vote (PMV) model from Fanger is the most widely used comfort model [36]. Generally, PMV is an index with a seven-point scale, where −3 is the cold extremum and +3 is hot. PMV model can be expressed as:

$$PMV = f (M, I, Ta, Tr, RH, v)$$

where,

M = metabolic rate;
I = clothing insulation;
Ta = Air temperature;
Tr = radiant temperature;
RH = Relative humidity;
v = Air velocity.

PMV can be directly connected with productivity level in an office while performing basic tasks like typing or thinking. Most of the PMV index studies are based on climate chambers [37], which can be very different from a normal office or home environment. Another alternative approach is adaptive comfort; thousands of building occupants have been involved in field studies in real buildings, where measurements and questionnaires have been used to correlate the temperature to the thermal sensation experienced by the occupants [37].

Air and radiant temperatures are critical parameters of thermal comfort in modern buildings, which could be maintained with the use of phase change materials. For example, using phase change materials in the roof and walls can reduce their interior temperature, which will have a direct effect on the convection (linked to the air temperature) and radiation (linked to the radiant temperature) heat transfer, and subsequently improve the thermal comfort of people inside the building. Phase change materials (PCMs) have great potentials to be used in modern building materials to stabilize indoor temperature fluctuations for improving thermal comfort by phase transition occurs between the "cooling" and "heating" temperatures and heat absorbed/emitted without the material changing temperature [38].

Aside from the thermal load of a building, another vital parameter is the thermal comfort of a building, and this is the condition of mind which expresses satisfaction with the thermal environment [39]. It means that a person feels neither too cold nor too warm. It is essential for health, well-being and productivity [40,41]. Thermal comfort is vital for buildings, especially in tropical climates where intensive cooling capacity is needed throughout the year [42]. Therefore, PCM is commonly investigated in passive cooling in buildings to provide thermal comfort by adding construction materials or integrating them into building structures [43]. In tropical regions such as Darwin in Australia, which is a warm, humid climate that requires long-term conditioning to gain thermal comfort, more heat is usually dissipated into the environment because a hotter environment raises the temperature and demands thermal comfort in the buildings [44].

Generally, it has been observed that PCM's efficacy depends on various parameters such as the type of PCM, location, climate, phase change temperature range, enthalpy curve, and thickness [45,46]. However, little study has been carried out on building orientation on PCM's melting temperature. Moreover, the overall efficacy of PCM in hot and humid climates is equally unclear [47]. Apart from these, there is no evidence of standard thickness of PCM's layer to optimize the thermal comfort inside a building in different climate zones [48]. Although some previous studies suggest that during the phase transition, a large amount of heat is absorbed and released by PCM, a phase change takes place in a specific temperature range rather than immediately at one temperature, for example, the specific heat capacity of PCM when used as a function of temperature [49].

The efficacy of PCM requires sizeable daily temperature variations to obtain melting-freezing cycles, but in Darwin, this could be a challenge [50]. Finally, various studies have been done in winter and summer climate regions (China, US, Europe, Bangladesh, India), but just a few studies (about 4%) have been published on tropical areas such as Darwin, Australia. Furthermore, just about 13% of review articles focused on the general problem of Thermal Energy Storage (TES) using PCM, 40% dealt with the development and evaluation of PCM in laboratories, 37% focused on numerical simulation, and 10% dealt with both experiment and simulation, while there are very few articles that focused on actual outdoor conditions [51].

Hence, this research aimed to study phase change materials to optimize indoor thermal comfort and minimize the cooling loads of residential buildings in tropical climates such as Darwin in Australia. The strategy used to achieve this was the incorporation of PCM-based gypsum wallboard on the walls of a single-family building in Wagaman, Darwin, using Energy Plus simulation software (Revit 2021). This work provides a combined modelling and simulation study to quantify the effectiveness of PCM in reducing cooling loads by design strategy of placement of PCM on the lining of the walls.

## 3. Materials and Methods

The approach used for this study consisted of a qualitative research mechanism that focused on Darwin's weather study, analysis of cooling loads, building modelling, PCM material selection and energy simulation.

### 3.1. Approach

This study was aimed at investigating phase change materials (using paraffin wax) to optimize indoor thermal comfort and minimize the cooling loads of residential buildings in tropical climates. Figure 2 illustrates the three-stage process that was adopted for this study. In the first stage, a single-family residential building was chosen and surveyed for this study.

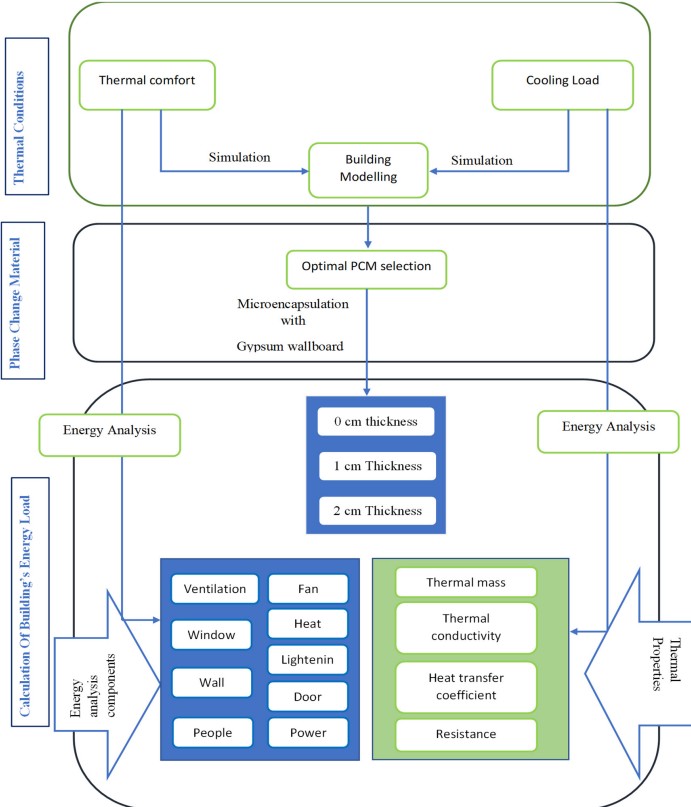

**Figure 2.** Adopted approach for calculation of the thermal comfort and cooling loads.

The orientation, 3-D model structure, surrounding microclimatic properties were studied by simulation using Revit 2021 software. In the second phase, 10% paraffin-based *n*-octadecane in pouches was used and mixed with gypsum to form the PCM-based gypsum wallboard of 1-centimetre and 2-centimetre-thicknesses.

In the final stage, simulation was used to calculate the cooling loads and analyse the energy consumption of the building. The simulation was performed in two stages. First, the construction of the building and secondly, the introduction of the data such as internal gains, environmental aspects and cooling system.

The cooling load associated with different building envelopes and the energy consumption of the selected building (based on a building's conceptual masses, building elements, or both depending on the analysis mode selected in the energy setting dialogue) was calculated using the Revit software and Energy Plus, respectively.

### 3.2. General Simulation Process

Revit 2021 software was used to structure a 3D model of the building located at Wagaman suburb in Darwin, Australia. The 3D model in Revit usually represents 3D geometry as rectangles, lines, and extrusion operations between the geometry. The material properties for the materials used in the construction of the building were conveniently obtained from the material properties browser of the Revit software. The analyse toolbar gave the option to calculate the cooling loads associated with different building envelopes. After modelling the building in Revit2021, the spaces were divided into zones to calculate the cooling load.

In terms of energy consumption analysis for the selected residential building, the project's geographic location, weather, and site were specified in Revit. Also, the cooling design temperature data for all months of the year as shown in Darwin were specified. Since the location, weather and site data were provided, the 3D structure was converted into the energy analytical model to calculate the yearly energy consumption of the building. The energy analytical model was validated based on conceptual masses and building elements before running the energy simulation using the selected energy mode in the energy setting dialogue.

### 3.3. Building Survey

A realistic estimation of the thermal load of a building was performed using a building survey which included surveying the orientation, use of spaces, dimensions of spaces, ceiling height, construction materials, surrounding conditions and details about the building envelopes [52–54]. A single-family residential building with four spaces was selected for this study. The orientation of the building was 35–180 degrees, as shown in Figure 3A. The window–wall ratio for southern, western, and eastern interacts with window properties to impact daylight, heating and cooling. The building's walls were constructed from composite masonry material, including different insulation layers such as the core boundary layer, membrane layer, thermal/air layer, finish layer, brick, and render beige. In addition, a finished surface (gypsum wallboard) was added to the walls, as shown in Figure 3B below.

The material properties of composite masonry brick walls and the material properties of the PCM are given in Table 1, while the details of the building's outdoor design conditions are given in Table 2.

The floor plan of the building consisted of 4 spaces, as shown in Figure 4.

The external structured wall (render on brick on block) consists of concrete masonry material with a nominal thickness of 16.42 cm (concrete masonry is 7.48, thermal air is 1.97 and thermal air cavity fill is 1.97, brick is 4.02 and finish 2 render smooth is 0.98 respectively) in which two leaves of undressed stones can be built together. The interior partition wall consists of gypsum with core boundary metal stud layer (135 mm partition (2-h) with total thickness 5.33 mm. The floor and roof structures are designed to represent real buildings with typical construction details as the floor has graphics parameters (callout tag) with a

callout head with a 3-millimetre corner radius and a sim reference label, whereas roof is gable type at 30 degrees inclination and floor to roof height is 2743 mm.

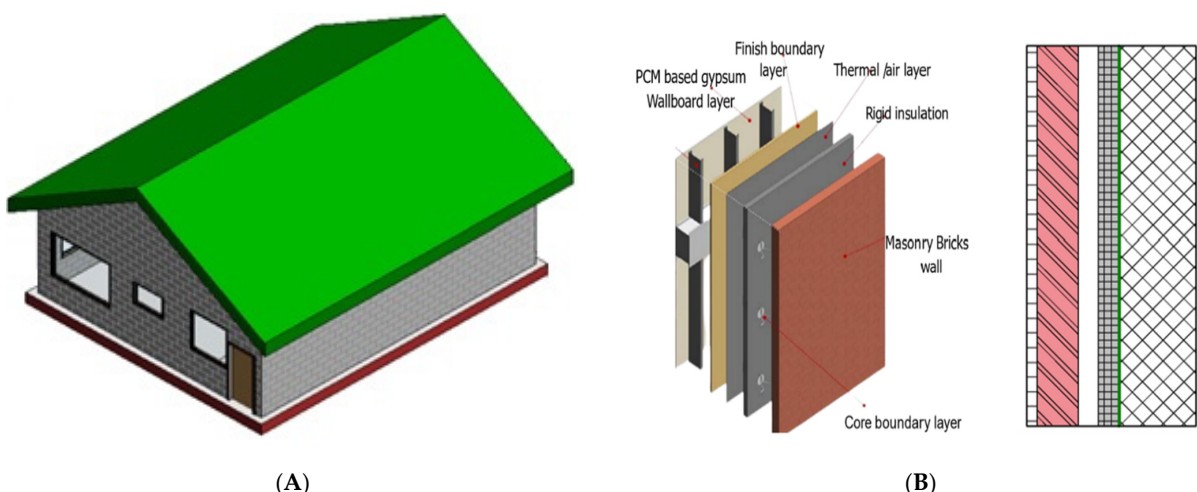

(**A**) (**B**)

**Figure 3.** (**A**) The 3D structure of the single-family residential building. (**B**) Composite masonry wall structure.

**Table 1.** PCM and composite wall's material properties.

| Composite Wall's Material Properties | | PCM (*n*-Octadecane) Material Properties | |
|---|---|---|---|
| **Parameters** | **Values** | **Parameters** | **Values** |
| Specific heat (J/kg/k) | 840 | Specific heat (J/kg/k) | 2150 |
| Density (kg/m$^3$) | 1800 | Density (kg/m$^3$) | 814 |
| Emissivity (W/m) | 0.95 | Emissivity (W/m) | 0.90 |
| Heat-transfer-coefficient U (W/(m$^2$·k) | 0.305 | Heat transfer coefficient U (W/(m$^2$·k) | 0.3059 |
| Thermal resistance (m$^2$·k)/W | 3.35 | Thermal resistance (m$^2$·k)/W | 3.268 |
| Thermal conductivity (w/(m·k)) | 1.3 | Thermal conductivity (w/(m·k)) | 0.650 |
| Absorptance | 0.7 | Absorptance | 0.7 |
| The thickness of the wall (cm) | 16.42 | Enthalpy (J/kg) | 245,000 |
| Thermal mass (kj/k) | 47.64 | Melting-temperature (°C) | 24 |

**Table 2.** Outdoor design aspects/conditions.

| Parameter | Value |
|---|---|
| Building type | Single-family |
| Location | 6/3 Carstens crescent, Wagaman, Darwin. |
| Area (m$^2$) | 83 |
| Volume (m$^3$) | 202.44 |
| Latitude (°) | −12.46 |
| Longitude (°) | 130.84 |
| Summer dry-bulb (°C) | 34 |
| Summer wet-bulb (°C) | 27 |
| Mean daily range (°C) | 6 |
| Design month | December |

*3.4. PCM Incorporation*

The method used to incorporate the PCM with gypsum was microencapsulation. This method prevented the external environment from contaminating the PCM. Tiny spheres of paraffin (5–10 microns in diameter) were encapsulated in acrylic shells, mixed with the gypsum in wallboard. Only 10% paraffin-based *n*-octadecane in pouches was used and mixed with gypsum to form the PCM-based gypsum wallboard. Additionally, to determine the maximum efficiency of the PCM-based gypsum wall, different thicknesses (1 cm and

2 cm) of PCM wallboard were applied at the internal and external sides of the composite masonry wall. The melting temperature of the PCM was set to 24 °C. The solidification temperature was assumed to be 1 K higher than the melting point of 25 °C. The phase change process inside the wallboard allowed a portion of the solar energy to be stored as latent heat [55].

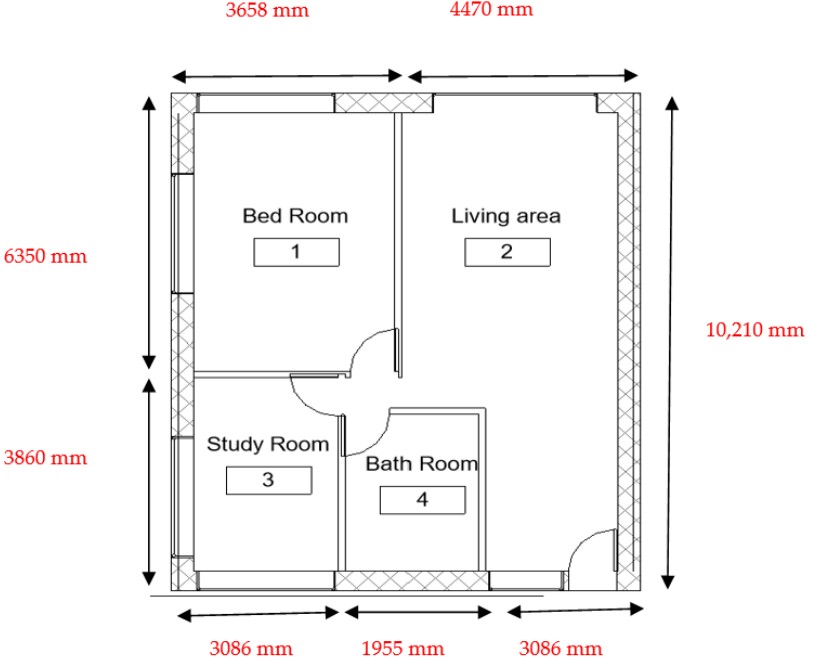

**Figure 4.** Floor plan of the building (10,210 mm × 8128 mm) dimensions and floor plan of house.

### 3.5. Design Conditions

Both indoor and outdoor design conditions were considered in calculating the cooling load of the building. Outdoor design conditions included climatic design information according to latitude, longitude, altitude and atmospheric pressure, statistical analysis of weather data, and data found in handbooks such as ASHRAE Fundamentals Handbook [32]. Indoor design conditions included basic design parameters such as thermal comfort and indoor air quality, solar gain through glass and structures, infiltration and ventilation, and internal heat gain. All indoor design conditions are represented in Table 3.

**Table 3.** Indoor design aspects/parameters.

| Internal Inputs | Values |
| --- | --- |
| Area ($m^2$) | 83 |
| Volume ($m^3$) | 202.44 |
| Cooling setpoint | 21 °C |
| Heating setpoint | 12 °C |
| Supply air temperature | 12 °C |
| Number of occupants | 1 |
| Infiltration | 0.0 |
| Air volume calculation type | VAV-Single Duct |
| Relative Humidity | 44.00% (calculated) |
| Thermal resistance ($m^2{\cdot}k$)/W | 3.35 |
| Building equipment | Concrete masonry |
| Indoor heat source | Water heaters |
| Cardinal orientation | 12.3810040, 130.8805245 |
| Cooling coil entering dry-bulb temperature | 28 °C |
| Cooling coil leaving dry-bulb temperature | 12 °C |
| Cooling coil entering wet-bulb temperature | 19 °C |
| Cooling coil leaving wet-bulb temperature | 14 °C |

### 3.6. Modelling of the Building

Revit 2021 software was used to structure a 3D model (Figure 5) of a building located at Wagaman suburb (GPS—12.3810040, 130.8805245) in Darwin, Australia. The 3D model in Revit usually represents 3D geometry as rectangles, lines, and extrusion operations between the geometry [56]. Depending on its purpose, the focus moves textured or shaded surfaces to volumes of building envelopes for calculations. With Revit, a building's 3D model's different sections and view can be seen straightforward from the wire-frame display. The Revit software was chosen because of its numerous advantages such as visualization, user-friendly, relative ease of understanding, showing depth elevation clearly and visually, and having the flexibility of design changes [56]. It helps to minimize the field clashes and delay in construction projects as resolution occurs before any work is done.

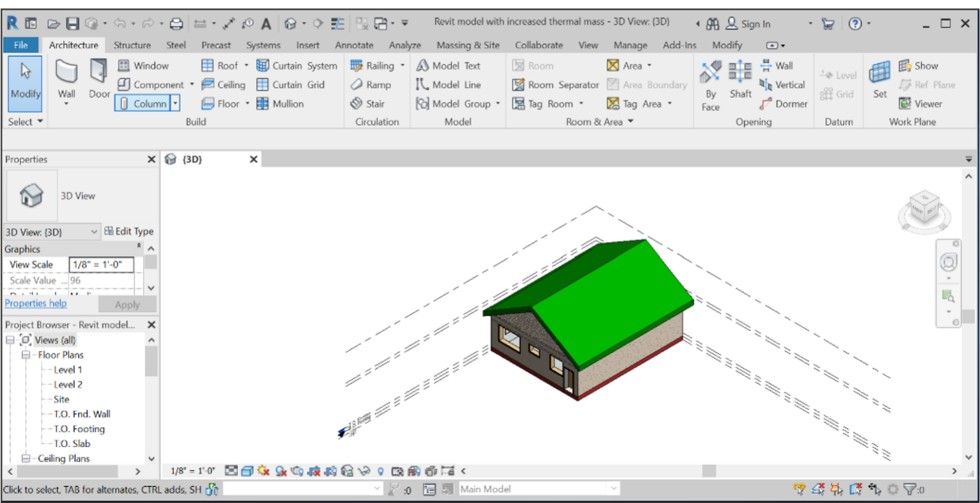

**Figure 5.** 3D models of the building from Revit 2021 software.

Furthermore, any changes in size or materials do not create any additional cost and the material properties for the materials used in the construction of the building were conveniently obtained from the material properties browser of the Revit software. In addition, the analyse toolbar gave the facility to calculate the cooling loads associated with different building envelopes. After modelling the building in Revit2021, the spaces were divided into zones to calculate the cooling load (Figure 6). Asides from the cooling load, the software also provided options for energy analysis in Analyze Tab Energy Optimization panel, based on a building's conceptual masses.

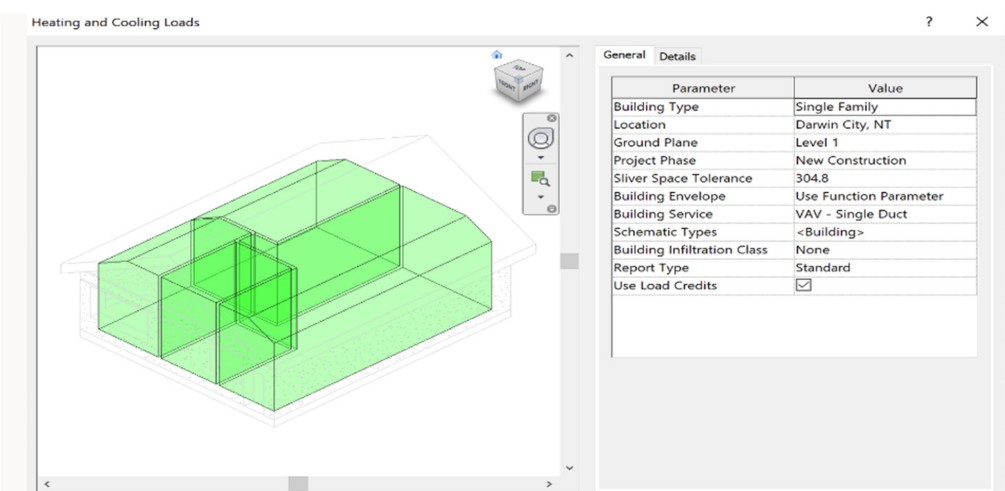

**Figure 6.** 3D model of Heating and cooling load zones and parameters from Revit.

### 3.7. Energy Consumption Analysis

To analyse the energy consumption for the selected residential building, Revit gave an option to specify the project's geographic location, weather, and site. In addition, the selection of Darwin, NT gave cooling design temperature data for all months of the year, as shown in Table 4. Since the location, weather and site data were provided, the 3D structure was converted into the energy analytical model to calculate the yearly energy consumption of the building. This feature allowed inspection of the energy analytical model to validate it before running the energy simulation. The energy analytical model was based on conceptual masses, building elements, or both depending on the analysis mode selected in the energy setting dialogue.

**Table 4.** Cooling design temperatures.

|  | January | February | March | April | May | June | July | August | September | October | November | December |
|---|---|---|---|---|---|---|---|---|---|---|---|---|
| Dry bulb (°C) | 33 | 33 | 33 | 34 | 33 | 33 | 33 | 33 | 33 | 34 | 34 | 34 |
| Wet bulb (°C) | 28 | 28 | 27 | 27 | 26 | 24 | 24 | 24 | 26 | 27 | 28 | 27 |
| Mean daily range (°C) | 6 | 6 | 6 | 8 | 9 | 9 | 10 | 10 | 8 | 7 | 7 | 6 |

### 3.8. Energy Simulation of the PCMs

In this study, energy simulation for the PCM was carried out using Energy Plus 9.5. The software tested the building energy performance using construction materials and the software equally had features that simulated the heating and cooling loads, cooling, lighting, ventilation, air conditioning, and energy flow [57]. The principal used parameters to determine the efficiency of phase change materials were material property, location, climate, sizing period and design day. Additional properties such as temperature-dependent thermal conductivity and enthalpy were used and solved by Heat Balance Algorithm (Conduction Finite Difference). The design day object in Energy Plus created the parameters for the program to create the 24-h weather profile that was used for sizing and running to test the other simulation parameters. Parameters in design day included a month and day; a day type consists of appropriate schedules for either sizing or simple tests, minimum or maximum temperature, wind speed and solar radiation values. In this study, simulation was performed in two stages. First, the construction of the building and secondly, the introduction of the data such as internal gains, environmental aspects and cooling system.

## 4. Results

### 4.1. Building Survey Result

The wall construction of the building represented the overall ability of wall constructions to resist heat losses and gains. The material of the walls was masonry. The roof construction represented the overall ability of roof constructions to resist heat losses/gains; the roof material was an uninsulated R60. The unintentional leaking of air into or out of the conditioned spaces called infiltration was 2.0 ACH-0.17 ACH, taken due to gaps in the building envelopes. The lighting efficiency was 20.4–3.23 W/m$^2$, representing the average internal heat gain and power consumption of electric lighting per unit floor area. The result of the building survey is summarised in Table 5 below.

### 4.2. PCM Incorporation/Microencapsulation

A detailed view of microencapsulated PCM is illustrated in Figure 7A along with the placement of gypsum wallboard at the interior of the composite masonry wall (Figure 7B) The graphical representation of the results of the effects of 1 centimetre- and 2 centimetre-thick PCM-based gypsum wallboard applied at the interior and the exterior surface of the wall is illustrated in Figure 7C,D, respectively.

### 4.3. Thermal Mass

The effects of varying thicknesses (1 cm and 2 cm thickness) of PCM-based gypsum incorporation on the thermal mass of the selected building in Wagaman, as well as without

PCM, are displayed in Figure 8A–D. It was observed that the thermal mass and resistance increased with an increase in PCM thickness, while the thermal conductivity and heat transfer coefficient decreased as the PCM-based gypsum wallboard thickness increased from 1 cm to 2 cm.

**Table 5.** Summary of building survey results.

| Parameter | Value |
|---|---|
| Wall material | Masonry |
| Roof material | Uninstalled R60 |
| Infiltration | 2.0 ACH-0.17 ACH |
| Calculated Results | |
| Power consumption | $3.23 \text{ W/m}^2$ |
| Average Internal heat gain | 20.4 |
| Peak cooling total load (W) | 21, 850 |
| Peak cooling sensible load (W) | 21, 088 |
| Peak cooling latent load (W) | 762 |
| Maximum cooling capacity (W) | 21, 850 |
| Peak cooling airflow (L/s) | 1, 134.4 |
| Peak heating load (W) | 2, 431 |
| Peak heating airflow (L/s) | 161.9 |
| Checksums | |
| Cooling load density ($\text{Wm}^2$) | 263.19 |
| Cooling flow density ($\text{L/s·m}^2$)) | 13.66 |
| Cooling flow/load (L/s.kw)) | 51.91 |
| Cooling area/load ($\text{m}^2/\text{kW}$) | 3.80 |
| Heating load density ($\text{W/m}^2$) | 29.28 |
| Heating flow density ($\text{L/(s·m}^2)$) | 1.95 |

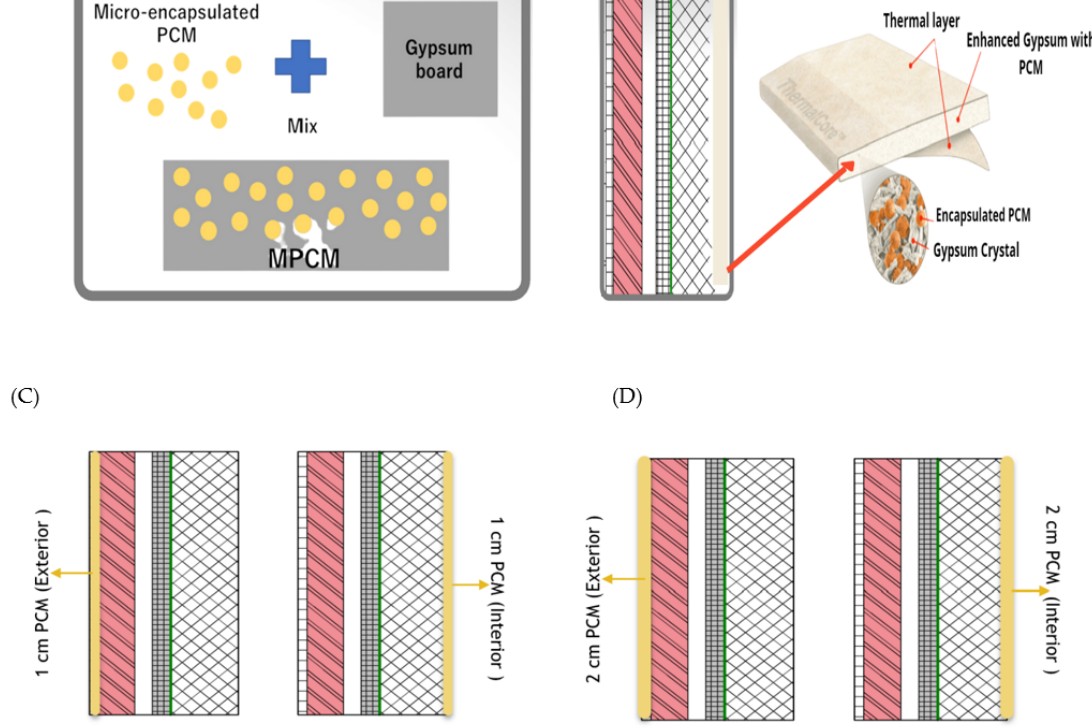

**Figure 7.** (**A**) Formation of microencapsulation based PCM. (**B**) Composite wall Microencapsulated PCM layer. (**C**) PCM-based gypsum wallboard with 1-centimetre thickness impregnated at exterior and interior side of the composite wall. (**D**) PCM-based gypsum wallboard with 2-centimetre-thickness impregnated at exterior and interior side of the composite wall.

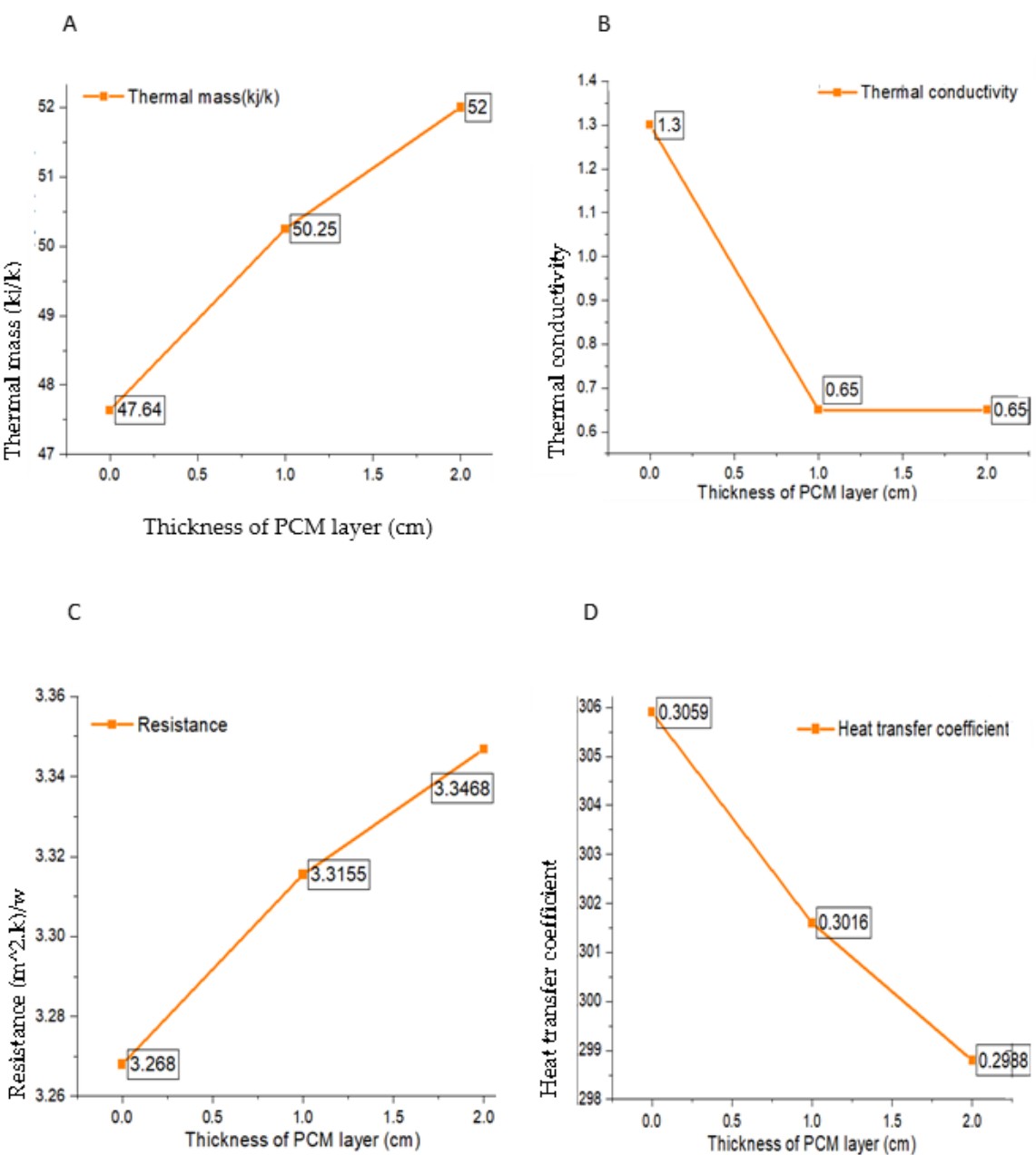

**Figure 8.** Impact of PCM thickness on thermal properties of the material, (**A**) impacts on building's thermal mass, (**B**) on thermal conductivity, (**C**) on building's resistance, and (**D**) on heat transfer coefficient.

*4.4. Effect of PCM Thickness Variation on Cooling Load*

The total cooling load of the building was calculated for both cases (with and without PCM incorporation). The maximum cooling load without PCM was 5362 W, and this was observed in the window component, while the door contributed the least (27 W). The results of the total cooling load of the building with composite masonry walls, including other components such as walls, windows, door roof and many others are shown in Figure 9A. With PCM incorporation (Figure 9B), the first calculation was done for 1-centimetre-thick PCM-based gypsum wallboard at the interior and exterior sides of the walls and the result showed a total cooling load at the exterior to be 7181 W, whereas PCM at the interior gave a total of 7176 W, as shown in Figure 9C. When a 2-centimetre PCM layer was applied at the internal and exterior of the wall, the total cooling loads were 7157 W, as shown in Figure 9D.

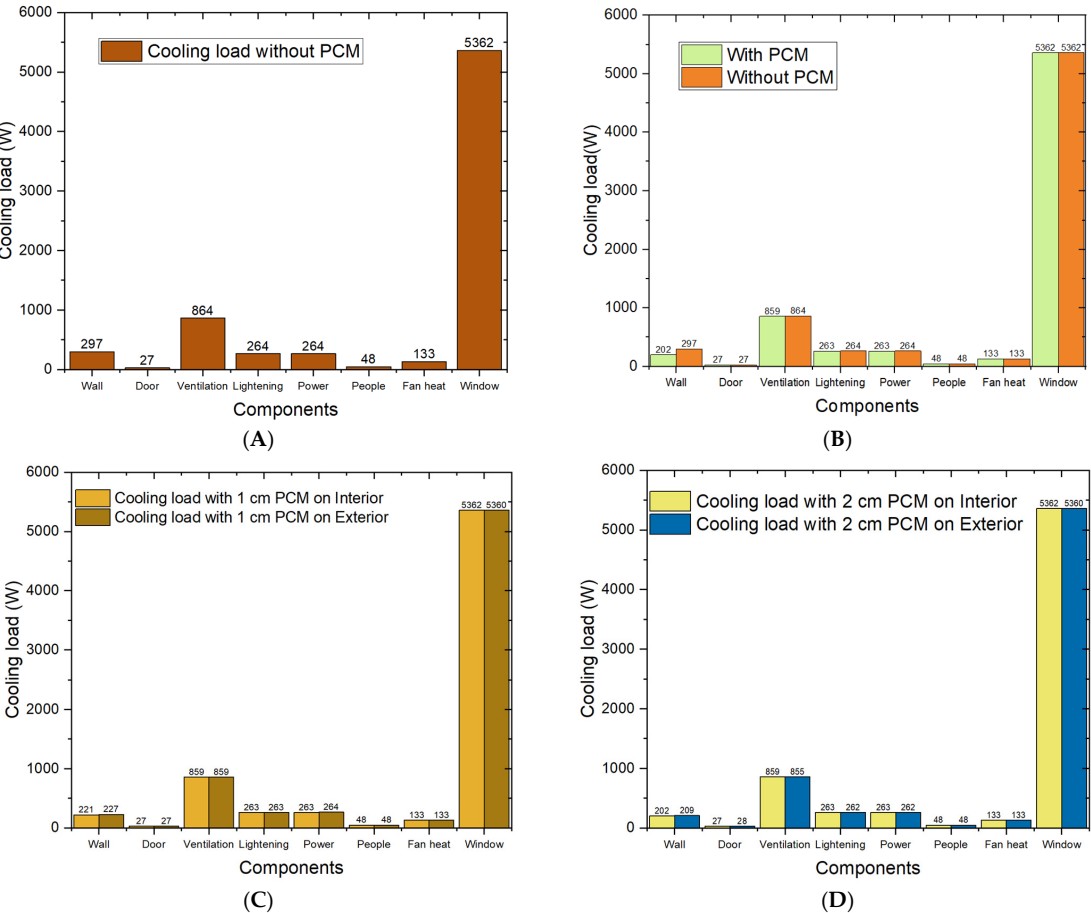

**Figure 9.** (**A**) Total cooling loads of the building with and without PCM. (**B**) Building's total cooling load without PCM at the exterior and the interior. (**C**) Building's total cooling load with 1-centimetre PCM layer. (**D**) Building's total cooling load with 2-centimetre PCM layer at the exterior and the interior.

This simulation focused majorly on the wall of the building, so for the composite masonry wall component only, the total cooling load of the building without PCM was 297 W. However, when 1-centimetre-thick PCM wallboard was applied at the internal side of the composite wall, the calculated cooling load was 221 W. The minimum cooling load was obtained when 2-centimetre-thick wallboard was applied at the wall; this time, the calculated load was 202 W. The graphical representation of total cooling loads due to composite masonry walls is represented in Figure 10.

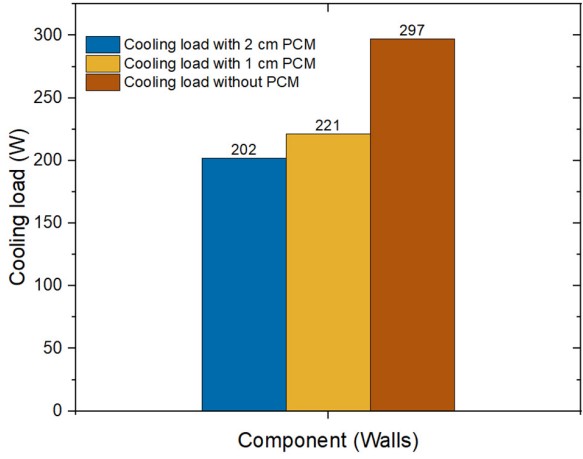

**Figure 10.** Cooling load effects on the masonry walls with PCM and without PCM.

### 4.5. Energy Consumption of the Building

The energy analysis gave the total site energy, source energy, and energy per conditioned building area with and without PCM are shown in Table 6.

**Table 6.** Building energy analysis with and without PCM (Energy Plus 9.5 version).

| Parameter | Without PCM | With 1 cm PCM | With 2 cm PCM |
| --- | --- | --- | --- |
| Total site energy (kBtu) | 172,042.20 | 159,481.39 | 159,047.8 |
| Total source energy (kBtu) | 242,830.08 | 232,296.90 | 229,262.72 |
| Energy per conditioned building area (kBtu/ft$^2$) | 205.57 | 189.86 | 187.34 |

The building's energy simulation was determined using Autodesk Insight 360, which gave the energy cost range and better presented the reality of the building's energy simulation, especially at conceptual design, and helped to compare actual building energy usage and savings. The Autodesk Insight 360 was comprised of a mean, minimum, and maximum energy cost automatically computed from a few dozen Department of Energy (DO22.2) (Building energy simulation and cost calculation software) and whole-building energy simulations.

The ECR was driven by individual building design and operation factors that influenced overall energy use. These factors included lighting efficiencies, roof and wall constructions, window glass, HVAC, and operating schedules.

Overall, the reduced cooling loads due to impregnation of the PCM-based gypsum wallboard resulted in reduced energy consumption. The energy analysis with a 1-centimetre-thick layer gave 172,042 kBtu total site energy, 242,830.08 kBtu source energy, and 205.57 kBtu energy per conditioned space without PCM, whereas, with a 2-centimetre-thick PCM layer, the energy analysis 159,047 kBtu total site energy, 229,262.72 kBtu total source energy and 187.34 kBtu for energy per conditioned space. The energy consumption cost with a 2-centimetre-thick PCM layer and without PCM is 18.0 USD/m$^2$/year and 18.9 USD/m$^2$/year, respectively. A graphical representation of energy analysis and cost is shown in Figure 11.

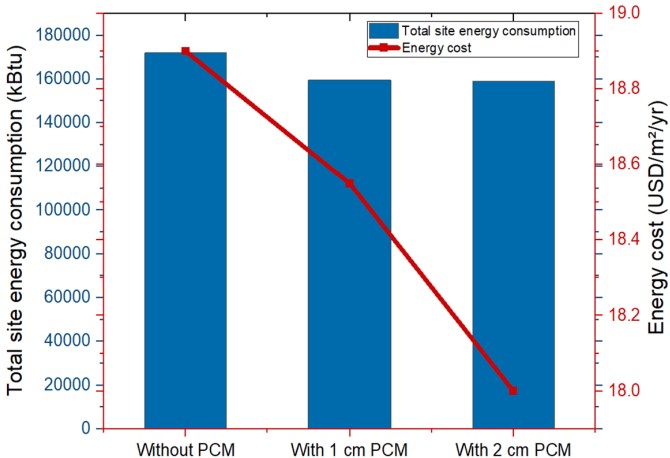

**Figure 11.** Total site energy consumption and energy cost with and without PCM.

## 5. Discussion

The average temperature of Darwin lies between 19 °C to 38 °C throughout the whole year. Despite this temperature range, the average relative humidity is approximately 38% to 75%. These weather patterns demand high conditioned space inside buildings to meet the required thermal comfort. Ultimately, it increases the energy consumption of buildings and cooling loads vice-versa. According to ASHRAE Standard 55-1992, the Human Comfort Zone diagrammed onto a Psychometric Chart lies between 24 °C (75.2 °F) to 27 °C (80.6 °F). However, Darwin's weather results compared to the ASHRAE Standard

55-1992 human comfort zone indicates that the indoor building environment of Darwin needs reduced temperature.

Thus, in this study, PCM was used to optimize the indoor thermal comfort and minimize the cooling loads using PCM-based gypsum wallboard which was incorporated into the walls of a single-family building in Wagaman, Darwin Australia.

### 5.1. Effect of PCM on the Total Cooling Load of the Building

Overall, the cooling load of the building with and without PCM was calculated and the total cooling load without PCM was 7259 W, whereas with PCM (1-centimetre and 2-centimetre-thicknesses), the total cooling load was 7151 W; this gave a 1.49% reduction in total cooling load. This decreased cooling load is due to the PCM-based gypsum wallboard that stored thermal heat due to the phase change mechanism. This slight cooling load percentage reduction could be increased by increasing the quantity of PCM during encapsulation, but the PCM that can be used is limited by the structural integrity of the gypsum board [58]. In addition, Darwin has a similar temperature during the day and night, thus restricting the PCM's ability to give an optimal performance [59].

### 5.2. Thermal Mass and Effect of PCM Thickness on Thermal Properties and Cooling Load

In this study, the thermal mass was developed by embedding PCM with gypsum wallboards and impregnated on the interior side of the walls. The PCM embedded gypsum wallboard used the energy associated with the material's phase change from solid to liquid. The PCM with a melting point of 24 °C added a significant equivalent thermal mass. Thus, embedded PCM specifically paraffin waxes can be designed with a melting point suitable for retrofitting in conventional buildings [60]. It was observed that the thermal mass of the walls increased with the PCM layer thickness increment, while thermal conductivity decreased with the PCM-based gypsum wallboard thickness increment. This increment in thermal mass increased the adequate heat capacity during the phase transition and stored more heat, consequently reducing the cooling load inside the space. Therefore, the use of PCM is an efficient passive measure to lower the cooling loads and cooling energy demands in buildings [61].

Usually, it is advantageous to use materials with low thermal conductivity to save building energy [62]. However, their low thermal conductivity can be a challenge since PCM stores large amounts of heat and cold in small volumes and because it is necessary to transfer this heat to the outside of the storage to use it. Consequently, there is a limitation in its use without instant heat transfer when required, no matter how large a capacity of heat storage the PCM has. These deficiencies could potentially reduce the rate of heat storage and release during the melting and solidification cycles and restrict their extensive applicability [63].

### 5.3. Effect on Cooling Load by Varying Thickness and Position of the PCM Layer

The total cooling load of the building decreased by increasing the thickness of the PCM-based gypsum wallboard. For example, the total cooling load decreased by 1.1% when 1 cm in thickness was applied to the wall, whereas a 1.5% reduction was obtained when 2 cm of PCM layer was applied. Therefore, a lower amount of energy to transfer indoors with increased thickness [64]. The highest reduction in total cooling load was obtained when PCM gypsum wallboard was installed at the internal linings of the walls and successful in capturing solar energy, which helps in performing the phase-change mechanism [58]. The total site energy and energy cost both decreased with increasing the PCM layer's thickness. Moreover, a 2-centimetre-thick PCM layer gave a 7.6% reduction in total site energy, and 4.76% energy saving in USD/m$^2$/year was achieved.

### 5.4. Thermal Resistance and Heat Transfer Coefficient

The thermal resistance of the wallboard increased with the increase in the thickness of the PCM layer. The higher thermal resistance of walls material was considered to restrict

heat transfer. Furthermore, due to the higher thermal resistance of PCM, heat transfer was lower towards the inside surface. This phenomenon indicated that the lesser the heat transfer to the indoor, the lesser cooling load would be required to maintain the thermal comfort. Additionally, the heat transfer coefficient decreased with an increase in PCM wallboard thickness. As mentioned in the introduction section, heat transfer through the wall of resistance is the reciprocal of heat transfer coefficient (U = 1/R), where R is the total resistance of the material used in constructing the walls. Therefore, the higher the R-value, the higher the insulating value of the material or the lower the U-value, the higher the insulation value of the material [32]. Hence, decreased value of the heat transfer coefficient gave higher insulation and less heat transfer to the indoor space.

### 5.5. Insights into Future Research

1. This study focused on the building walls component, which contributes less percentage toward the total cooling load of a building; however, the study on other parameters such as windows and roofs can provide more efficient results as they contribute more towards building total cooling load thus, additional research is required in this area.
2. Additionally, a small percentage of Cooling load reduction was obtained with organic PCM-based gypsum wallboard during this study, however, other PCM materials could be used for maximum results.
3. This research is simulation-based; some of the parameters are by default in Energy Plus software which cannot be changed; however, in a real-life experiment, we can use the actual parameters such as ground surface temperature to analyze the performance of PCMs.
4. In this study, the microencapsulation method was used to apply to the PCM. This is expansive, so other methods such as direct incorporation, shape stabilization could be tried for better performance and energy savings.

### 6. Conclusions

This study investigated the incorporation of PCMs based gypsum wallboard at the internal side of composite walls of a building located in Darwin, Australia. The studied wall was composed of bricks, air/thermal and rigid insulation. The effect of PCM position and thickness was assessed in two scenarios. Firstly, the PCM thickness was taken and 1 cm was applied to the interior/exterior surface of the wall. Secondly, 2-centimetre-thick PCM was applied at the interior/exterior. Hence, increased PCM thickness gave reduced cooling load and energy consumption. In addition, the following can also be deduced from this study on the thermal performance of the wall with PCM:

1. Additional thermal was achieved by applying a PCM impregnated gypsum board.
2. A reduction of 1.49% in total cooling load was obtained due to the lower thermal conductivity of the phase change material.
3. The maximum reduction in the building's total cooling load and site energy was obtained when PCM gypsum wallboard was installed at the internal side of the walls.
4. A 2-centimetre-thick layer of PCM wallboard gave a maximum reduction in cooling load when compared to 1 cm.
5. PCM optimum position and thickness gave a 7.6% reduction in total site energy and 4.76% energy saving in USD/m$^2$/year.
6. With the effective building envelope design and simulation results, the dependency on active means of a mechanical system was reduced.
7. In summary, our result showed that a 2-centimetre-thick layer of PCM wallboard gave a maximum reduction in cooling load when compared to 1 cm. This implied that heat gain could be reduced by 1.5 per cent through building envelopes by using 2-centimetre-thick PCM due to the lower thermal conductivity of the phase change material. The study also revealed that the location of PCMs with appropriate phase change temperature plays a vital role in solidifying the PCM layer. Furthermore, the

parametric results from the study showed that PCMs could give better performance when applied on an exterior wall than the interior wall because of PCM solidification at night times in tropical climates.

**Author Contributions:** Conceptualization, P.S. and H.M.-R.; methodology, P.S.; software, P.S.; validation, P.S. and H.M.-R.; formal analysis, P.S. and R.C.N.; data curation, M.A.U.R.T.; writing—original draft preparation, P.S. and R.C.N.; writing—review and editing, M.A.U.R.T.; visualization, P.S. and H.M.-R.; supervision, H.M.-R. and A.W.M.N. All authors have read and agreed to the published version of the manuscript.

**Funding:** This research received no external funding.

**Institutional Review Board Statement:** Not applicable.

**Informed Consent Statement:** Not applicable.

**Conflicts of Interest:** The authors declare no conflict of interest.

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
