# Peer review of "Performance Evaluation of Phase Change Materials to Reduce the Cooling Load of Buildings in a Tropical Climate"

_sustainability, doi:10.3390/su14063171_

Round 1

Reviewer 1 Report

The topic of the present study is interesting and globally relevant. It presents the potential for energy savings in the operation of energy systems in buildings, improving human thermal comfort in the indoor environment and reducing the environmental load. The authors investigated the cooling load of the building using computer simulations. Their work and effort is highly appreciated and without doubt the results of this research are of importance not only for the development in this research area but also for professional practice in the implementation of new modern energy efficient buildings.

The presentation of the research results in this study needs, in my opinion, additions and modifications to improve the form of the study and to explain in more detail the progress of the work and the outputs of the research and computer simulations. Therefore, I offer the following comments and suggestions for additions and modifications to the article:

  • In section 2. (Phase Change Materials) on page 2, it would be useful to add a table of the PCM distribution. PCMs are the basis of this study and therefore this section should be supplemented with important information and research results from this area. The selected paraffinic PCMs should be described in more detail (chemical composition, basic chemical equations for the reactions, graphs expressing melting temperature and enthalpy of phase change, etc.).
  • In section 2.1. (Heat transfer through a wall) on page 3, give the basic procedure for calculating heat transfer through building structures, not just the resulting equation for the wall. In what physical units is Q? Under equation (1), list each physical quantity with units. Consider whether to give the British units in European format or at least give the conversion, e.g. 1 kWh ≈ 3,412.14 Btu. State and describe the meaning of CLTD in equation (1).
  • In section 2.2. (Cooling load principle) on page 3, state the basic procedure for calculating the cooling load. A description without a calculation procedure is too general. Reference to the software is not sufficient.
  • In section 2.3. (Thermal comfort) on page 3, provide the basic parameters, equations and methods for assessing thermal comfort in buildings. Indicate which parameters of thermal comfort in buildings and how they can be influenced by the use of the PCM.
  • In section 3. (Materials and Methods) on page 4, the approach adopted to calculate the thermal comfort and cooling loads as shown in Figure 1 needs to be described in more detail in words. For example, divide the description into 3 stages and describe what will be implemented in which stage and how the stages relate and build on each other.
  • In section 3.1. (Building survey) on page 5 in the description of the building, also in Figures 2 and 3, the dimensions of the building, the composition of the structures with the thicknesses of the individual layers should be added. In Table 1 on page 6, correct the units of physical quantities. In rows 192, 197, 202, 203, 206 there is: 'Error! Reference source not found".
  • From section 3.2. (PCM incorporation) on page 7, it is not clear whether the gypsum plasterboards were also actually created or whether this model was predicted only for computer simulations. This needs to be explained. A picture would also help, with a real model a photograph would be helpful.
  • In section 3.3. (Design Conditions) on page 7, all design internal and external design conditions should be listed (quantified) specifically in the form of a clear table.
  • In section 3.4. (Modelling of the building) on page 7, provide images of 3D models of the building, for example a print screen from Revit 2021 software, and use them to describe the process of running the simulations.
  • In sections 3.5 (Energy consumption analysis) and 3.6 (Energy simulation of the PCMs) on page 8, it is important to quantify all the input data of the building under study, e.g. thermal resistances of the building structures, areas of the individual building constructions, indoor and outdoor temperatures, number of occupants, building equipment with indoor heat sources, orientation to the cardinal directions, heat losses of rooms and the building, heat gains of rooms and the building, ... Line 257 reads: 'Error! Reference source not found'.
  • The whole Chapter 4 (Results) on pages 8 to 16 needs to be edited, supplemented and also clear tables with the simulation outputs, for example cooling loads at different external conditions for the different building envelope walls studied, need to be included. Some of the figures are again numbered 1. It is necessary to describe all figures in detail in the text (or even directly in the figures). It is understandable that the authors are clear about what the figures show, but the reader must learn from the text what dependencies the figures show, what outputs (values of physical quantities) are presented, what is the meaning of the results, what are the individual values of the results compared to, what correlation do they have with each other, and so on. There are no references to figures and tables in the text, which makes the description of the simulation and research results even more incomprehensible. In the text and in the pictures it says: "Error! Reference source not found".
  • Chapter 5. (Discussion) on pages 16 to 18 also needs to be amended and supplemented. Each section needs to quantify the findings from the simulations and research as in, for example, Section 5.1 (Effect of PCM on the total cooling load of the building) and 5.3 (Effect on Cooling load by varying thickness and position of the PCM layer) on page 17. Sections 5.2, 5.4 and 5.5 need to be supplemented with specific simulation and research results.
  • In section 6. (Conclusions) on pages 18 and 19, please add what you consider to be your contribution to this research area and to professional practice in the construction of new energy-efficient buildings.

The authors of the present study have carried out a great deal of high-quality scientific work, for which I express my gratitude and my admiration. After incorporating my comments, editing and supplementing this article, I recommend its publication.

Reviewer 2 Report

Please, find out the attachment to perform the items of comments and suggestions.

Reviewer 3 Report

Not a bad article, it is even written well. Calculation of non-parafin capsuls is interesting. On the other hand - the article does not belong to a "Star publications"  - it is one many and many software simulations - in this case with Revit soft. So nothing special value or idea is here. But it can be published like it is, except for chaos with unit. 

Formal problems : Units you are mixing British and SI units, this should be corrected. (e.g. lines 111 or 352)

Round 2

Reviewer 2 Report

I believe that the experimental results obtained in this article are helpful for future extended research, and are also provided to the building materials application industry as a reference.